# Chronic Fatigue, Depression and Anxiety Symptoms in Long COVID Are Strongly Predicted by Neuroimmune and Neuro-Oxidative Pathways Which Are Caused by the Inflammation during Acute Infection

**DOI:** 10.3390/jcm12020511

**Published:** 2023-01-08

**Authors:** Hussein Kadhem Al-Hakeim, Haneen Tahseen Al-Rubaye, Abbas F. Almulla, Dhurgham Shihab Al-Hadrawi, Michael Maes

**Affiliations:** 1Department of Chemistry, College of Science, University of Kufa, Kufa 54002, Iraq; 2College of Medical Laboratory Techniques, Imam Ja’afar Al-Sadiq University, Najaf 54001, Iraq; 3Department of Psychiatry, Faculty of Medicine, Chulalongkorn University, Bangkok 10330, Thailand; 4Medical Laboratory Technology Department, College of Medical Technology, The Islamic University, Najaf 54001, Iraq; 5Al-Najaf Center for Cardiac Surgery and Transcatheter Therapy, Najaf 54001, Iraq; 6Department of Psychiatry, Medical University of Plovdiv, 4000 Plovdiv, Bulgaria; 7IMPACT, The Institute for Mental and Physical Health and Clinical Translation, School of Medicine, Barwon Health, Deakin University, Geelong, VIC 3220, Australia

**Keywords:** long COVID, depression, chronic fatigue syndrome, neuro-immune, inflammation, oxidative stress, antioxidants

## Abstract

Background: Long-term coronavirus disease 2019 (long COVID) is associated with physio-somatic (chronic fatigue syndrome and somatic symptoms) and affective (depression and anxiety) symptoms. The severity of the long COVID physio-affective phenome is largely predicted by increased peak body temperature (BT) and lowered oxygen saturation (SpO2) during the acute infectious phase. This study aims to delineate whether the association of BT and SpO2 during the acute phase and the long COVID physio-affective phenome is mediated by neurotoxicity (NT) resulting from activated immune-inflammatory and oxidative stress pathways. Methods: We recruited 86 patients with long COVID (3–4 months after the acute phase) and 39 healthy controls and assessed serum C-reactive protein (CRP), caspase 1, interleukin (IL) 1β, IL-18, IL-10, myeloperoxidase (MPO), advanced oxidation protein products (AOPPs), total antioxidant capacity (TAC), and calcium (Ca), as well as peak BT and SpO2 during the acute phase. Results: Cluster analysis revealed that a significant part (34.9%) of long COVID patients (n = 30) show a highly elevated NT index as computed based on IL-1β, IL-18, caspase 1, CRP, MPO, and AOPPs. Partial least squares analysis showed that 61.6% of the variance in the physio-affective phenome of long COVID could be explained by the NT index, lowered Ca, and peak BT/SpO2 in the acute phase and prior vaccinations with AstraZeneca or Pfizer. The most important predictors of the physio-affective phenome are Ca, CRP, IL-1β, AOPPs, and MPO. Conclusion: The infection–immune–inflammatory core of acute COVID-19 strongly predicts the development of physio-affective symptoms 3–4 months later, and these effects are partly mediated by neuro-immune and neuro-oxidative pathways.

## 1. Introduction

While the infection rate for severe acute respiratory syndrome coronavirus 2 (SARS-CoV-2) has fallen globally [1,2], a new concern has arisen among many post-infection individuals with long COVID symptoms [3]. Several studies have reported a cluster of persistent symptoms extending beyond full recovery for coronavirus disease (COVID-19) [4,5,6]. These symptoms appear after up to two (infection-related), three (acute post-COVID), six (prolonged post-COVID), or more (chronic post-COVID) months following the acute infectious phase [7]. Many individuals (74-87.4%) with long COVID suffer from a variety of mental and physio-somatic symptoms after recovery from the acute phase [8,9], including chronic fatigue, affective symptoms (low mood and anxiety), cognitive dysfunctions, and sleep disturbances, along with somatic manifestations, such as autonomic symptoms, muscle pain, muscle tension, headache, a flu-like malaise, gastro-intestinal symptoms (GISs), shortening of breath, persistent cough, and chest pain [5,6,8,10,11,12,13,14,15,16,17,18,19]. Available treatments are limited, as there is no clear understanding of the pathophysiological mechanism.

The immunopathogenesis of the acute infectious phase of COVID-19 comprises activation of the cytokine network with induction of interferons; interleukin (IL) 1, IL-6, IL-12, IL-18, and tumor necrosis factor (TNF) α signaling; activation of the nucleotide-binding domain, leucine-rich repeat, and pyrin domain-containing protein 3 (NLRP3) inflammasome; and activation of key antiviral pathways, toll-like-receptor cascades, and NOD-like receptor signaling [20,21,22,23]. The NLRP3 is a key component of the innate immune system and an intracellular sensor that is induced by pathogen-associated and damage-associated molecular patterns [24,25]. NLRP3 activation causes increased levels of caspase 1, IL-1β, and IL-18 and cell death or pyroptosis [26,27]. Increased peak body temperature (BT) and lowered oxygen saturation (SpO2) reflect the severity of the immune–inflammatory response during acute COVID and predict critical COVID-19 and increased mortality [28,29,30].

Recently, we introduced a new concept—namely, the physio-affective phenome of acute and long COVID—to describe the physio-somatic (fatigue and somatic symptoms) and affective (depression and anxiety) symptoms in both illnesses (see [29,31,32] for reviews). Thus, in acute and long COVID, a validated latent construct could be derived from depressive, anxiety, chronic fatigue, and multiple physio-somatic symptoms; therefore, this factor score reflects the severity of the intertwined increases in physio-somatic and affective symptoms [29,31,32].

In acute COVID-19, this physio-affective phenome was largely explained by the cumulative effects of increased pro-inflammatory cytokines, including IL-6 and soluble advanced glycation products (sRAGEs); changes in acute-phase proteins, including C-reactive protein (CRP) and albumin; lowered calcium (Ca); pneumonia as indicated by chest computerized tomography scan abnormalities (CCTAs); and diminished SpO2 [31]. Moreover, elevated peak BT and lowered SpO2 during the acute phase of infection predict the physio-affective phenome of long COVID [29]. In long COVID, the severity of the physio-affective phenome was also predicted by increased oxidative toxicity as indicated by increased levels of malondialdehyde (MDA), protein carbonyls (PCs), myeloperoxidase (MPO), and nitric oxide (NO), as well as lowered antioxidant defenses as indicated by lowered zinc and glutathione peroxidase (Gpx) [32]. Moreover, the oxidative toxicity/antioxidant ratio was significantly predicted by increased peak BT and lowered SpO2 [32]. These results indicate that the immune–inflammatory response in acute COVID-19 predicts, at least in part, the physio-affective symptoms of long COVID and that these effects are partly mediated by increased neuro-oxidative toxicity [31,32].

It is important to note that neuro-immune and nitro-oxidative toxicity play key roles in the pathophysiology of chronic fatigue syndrome (CFS), major depression (MDD), and generalized anxiety disorder (GAD) [33,34,35,36]. There is now also evidence that upregulation of the NLRP3 inflammasome plays a key role in MDD [37,38], fatigue [39,40], cognitive impairments [41], and anxiety [42,43]. Nevertheless, there are no data showing whether the NLRP3 inflammasome and lowered Ca are involved in the physio-affective phenome of long COVID.

Hence, this study was performed to determine (a) the effects of the NLRP3 inflammasome (IL-1β, IL-18, and caspase 1), inflammatory response (CRP), oxidative stress (advanced oxidation protein products (AOPPs) and MPO), total antioxidant capacity (TAC), and lowered Ca on the physio-affective phenome of long COVID; and (b) whether neuro-immune and neuro-oxidative toxicity is predicted by lowered SpO2 and increased peak BT during the acute phase. The specific hypotheses were that the physio-affective phenome of long COVID is predicted by lowered SpO2 and increased peak BT and that these effects are, at least in part, mediated by increased neurotoxicity (NT) due to increased caspase 1, IL-1β, IL-18, CRP, MPO, and AOPP and lowered TAC and Ca levels. The data were analyzed using the new precision nomothetic psychiatry approach [30,44], which involved the construction of a new endophenotype class of long COVID.

## 2. Participants and Methods

### 2.1. Participants

We used the World Health Organization (WHO, Geneva, Switzerland) criteria to diagnose long COVID patients [45]. These criteria were that: (a) daily life activities of the patients should be influenced minimally by two symptoms (fatigue, impairment of memory or concentration, achy muscles, absence of smell or taste senses, affective symptoms, and cognitive impairment); (b) patients should have had a confirmed infection with COVID-19; (c) the symptoms should persist beyond the acute phase or should become apparent 2–3 months later; and (d) the symptoms should last for at least two months [45] and be present 3–4 months after recovery. Accordingly, 86 long COVID patients participated in the current study, clustered into two groups: long COVID with low (n = 56) and high (n = 30) NT (as defined below). All Long COVID patients sought professional assistance for post-COVID symptoms and exhibited varying degrees of reduced functioning and productivity, as well as an increased inability to perform daily tasks as effectively as they did prior to COVID infection. Additionally, we recruited 39 apparently healthy controls who did not show any clinical signs of infection and showed negative test results in reverse transcription real-time polymerase chain reaction (rRT-PCR) tests when included in the study. All participants were recruited between September and December 2021. The present study involved a combined methodology; namely, investigation of the impact of acute-phase COVID-19 on long COVID symptoms (retrospective study design) and comparison of abnormalities in long COVID patients versus healthy controls (case–control design).

In the acute COVID-19 phase, specialized clinicians and virologists diagnosed patients with a positive COVID-19 infection based on: (a) severe symptoms of infection, such as fever, cough, shortness of breath, and the loss of smell and taste senses; (b) positive rRT-PCR test results; and (c) positive immunoglobulin M (IgM) against SARS-CoV-2. All patients were quarantined and treated at several hospitals and specialized centers within Al-Najaf city, Iraq: Al-Sader Medical City of Najaf, Al-Hakeem General Hospital, Al-Zahraa Teaching Hospital for Maternity and Pediatrics, Imam Sajjad Hospital, Hassan Halos Al-Hatmy Hospital for Transmitted Diseases, the Middle Euphrates Center for Cancer, and Al-Najaf teaching hospital.

All participants with long COVID were free of any signs of acute COVID-19—namely, dry cough, sore throat, shortness of breath, fever, night sweats, and chills—and, before participating in the current study, patients and controls showed negative rRT-PCR test results. Around 33% of the participants in the control group had minor mental symptoms, such as low mood, anxiety, and fatigue, resulting from the quarantine period and lack of social activities, which may also affect patients with long COVID. We excluded subjects with a previous major depressive episode, bipolar disorder, dysthymia, GAD, panic disorder, schizo-affective disorder, schizophrenia, psycho-organic syndrome, substance use disorders (except tobacco use disorder (TUD)), or neurodegenerative and inflammatory diseases, such as CFS [46]. We also excluded subjects with liver and renal disease, Parkinson’s or Alzheimer’s disease, multiple sclerosis, stroke, or systemic (auto)immune diseases, such as diabetes mellitus, psoriasis, rheumatoid arthritis, inflammatory bowel disease, and scleroderma, as well as pregnant and breastfeeding women.

The present study was designed and performed in line with Iraqi and international ethical and privacy laws, including the World Medical Association’s Declaration of Helsinki, the Belmont Report, the Council for International Organizations of Medical Sciences (CIOMS) Guideline, and the International Conference on Harmonization of Good Clinical Practice. Our institutional review board adheres to the International Guidelines for Human Research Safety (ICH-GCP). We obtained written consent from all participants, parents, or any legally responsible person prior to involvement in our study. The institutional ethics board and the Najaf Health Directorate—Training and Human Development Center approved our research according to their documents with the numbers 8241/2021 and 18378/2021, respectively.

### 2.2. Clinical Measurements

Three to four months after recovery from acute COVID-19, a semi-structured interview was conducted by a senior psychiatrist to obtain the socio-demographic and clinical characteristics of all participants. The psychiatrist assessed several symptom domains, including chronic fatigue and fibromyalgia symptoms utilizing the Fibro Fatigue (FF) scale [47], severity of depression using the Hamilton Depression Rating Scale (HAMD) [48] and the Beck Depression Inventory II (BDI-II) [49], and severity of anxiety symptoms using the Hamilton Anxiety Rating Scale (HAMA) [50]. Moreover, we utilized those rating scale items to derive subdomains for the major symptoms. We made two subdomains with the HAMD; the first was pure depressive symptoms (pure HAMD), which was the sum of sad mood, feelings of guilt, suicidal thoughts, and loss of interest; and the second was physio-somatic HAMD (physiosom HAMD), which was the sum of somatic anxiety, gastrointestinal (GIS) anxiety, genitourinary anxiety, and hypochondriasis. Likewise, two subdomains of the HAMA were computed; namely, pure anxiety symptoms (pure HAMA), the sum of anxious mood, tension, fears, anxiety, and anxious behavior during the interview, and physio-somatic HAMA symptoms (physiosom HAMA), the sum of somatic sensory, cardiovascular, genitourinary, and autonomic symptoms, as well as GIS. Furthermore, after omitting items reflecting cognitive and affective symptoms in the FF scale, a pure physio-somatic FF (pure FF) score was computed as the sum of muscular pain, muscle tension, fatigue, autonomous symptoms, GIS, headache, and flu-like malaise. We also computed the sum of all pure depressive BDI-II (pure BDI) symptoms—thus excluding physio-somatic symptoms—including sadness, discouragement about the future, feeling like a failure, dissatisfaction, feeling guilty, feeling punished, self-disappointment, self-criticism, suicidal ideation, crying, loss of interest, difficulties with decisions, and work inhibition. In our previous studies, the physio-affective phenome was defined [29,31] as the first factor extracted from the pure FF and BDI scores and the pure and physiosom HAMA and HAMD scores. The DSM-5 criteria were used to diagnose TUD. Weight in kilograms was divided by height in meters and squared to calculate the body mass index (BMI).

We used the patients’ records to determine the lowest SpO2 and peak BT values, which were obtained during hospitalization for the acute infectious phase. The assessments were undertaken by a well-trained paramedical professional who employed an electronic oximeter manufactured by Shenzhen Jumper Medical Equipment Co. Ltd. and a sublingual digital thermometer with a beep sound. By subtracting the z-transformed SpO2 values from the z-transformed peak BT values, we generated a new index involving both lowered SpO2 and increased peak BT (dubbed the TO_2_ index). Additionally, we recorded the different types of vaccines received by the patients; namely, AstraZeneca (Cambridge, UK), Pfizer (New York, NY, USA), and Sinopharm (Beijing, China).

### 2.3. Assays

Five milliliters of venous blood were sampled using disposable syringes at 7:30–9:00 am after an overnight fast. The blood was then transferred directly to serum tubes. We avoided any hemolyzed, lipemic, and icteric blood samples. All tubes were centrifuged at 3000 rpm after 10 min incubation at room temperature. Then, we produced three aliquots of serum, which were stored at −70 °C in Eppendorf tubes until they were thawed for assaying. We employed ELISA kits provided by Nanjing Pars Biochem Co., Ltd. (Nanjing, China) to assess serum levels of IL-1β, IL-18, IL-10, caspase 1, MPO, TAC, and AOPPs (albumin ratio). Total serum Ca was measured spectrophotometrically using ready-to-use kits obtained from Agappe Diagnostics Ltd., Cham, Switzerland. We used a z unit-based composite score to determine a new NT index by computing the z transformation of IL-1β (z IL-1β) + z IL-18 + z caspase 1 + z MPO + z AOPP + z CRP.

### 2.4. Statistical Analysis

In the present study, IBM SPSS software, version 28, was used to carry out the statistical analyses. We conducted analysis of variance (ANOVA) to delineate the differences in continuous variables among the study groups and analysis of contingency tables to examine associations between nominal variables. Pearson product–moment correlation coefficients were used to analyze the relationship between two scale variables. Multivariate and univariate general linear models (GLM) were used to examine the association between clinical and biomarker data and the diagnostic categories while allowing or controlling for age, TUD, sex, BMI, and education. Estimated marginal mean (SE) values were computed and multiple group mean differences were assessed using Fisher’s protected (the omnibus test is significant) least significant difference (LSD). The ability of biomarkers and clinical variables to predict the physio-affective symptoms was determined through multiple regression analysis. We utilized an automated stepwise approach with a p-to-enter of 0.05 and p-to-remove of 0.06. For each of the explanatory variables, we computed the standardized beta-coefficients, t-statistics, and exact *p*-value, along with F-statistics and the total variance explained (R^2^). Furthermore, we used the variance inflation factor and tolerance to examine multicollinearity. The heteroskedasticity was checked by employing the White and modified Breusch–Pagan tests. We used cluster analysis (two-step) and followed the precision nomothetic approach [30] to construct endophenotype classes of patients with long COVID based on a combination of SpO2, peak BT, and the neurotoxic biomarkers in a z unit-based composite score (z IL-1β + z IL-18 + z caspase 1 + z MPO + z AOPP + z CRP) dubbed the NT index. The cluster solution was considered adequate when the silhouette measure of cohesion and separation was > 0.5. Canonical correlation analysis was employed to investigate the correlations between two sets of variables; namely, physio-affective symptoms 3–4 months after acute COVID infection as the dependent variables and biomarkers as explanatory variables. The variance explained by the canonical variables of both sets was computed, as well as the variance in the canonical dependent variable set explained by the independent canonical variable set. We accepted the canonical components when the explained variance of both sets was > 0.50 and when all canonical loadings were >0.5.

Partial least squares (PLS) analysis was used to study the causative relationships between SARS-CoV-2 infection, peak BT, and lowest SpO2 during the acute phase of disease and the physio-affective phenome of long COVID, whereby the effects of the input variables were partly mediated by NT and other biomarkers. All input variables were inputted as single indicators, and the output variable was a latent vector extracted from the values for the pure and physiosom HAMA and HAMD and pure FF and BDI (the physio-affective phenome). Complete PLS analysis was conducted only when the outer and inner models met the following prespecified quality criteria: (a) all loadings on the extracted latent vector were >0.6 at *p* < 0.001; (b) the output latent vector showed high construct and convergence validity, as indicated by rho A > 0.8, Cronbach’s alpha > 0.7, composite reliability > 0.7, and average variance extracted (AVE) > 0.5; (c) blindfolding demonstrated that the construct’s cross-validated redundancy was sufficient; (d) confirmatory tetrad analysis (CTA) demonstrated that the latent vector extracted from the rating scale scores was not mis-specified as a reflective model; (e) the model’s prediction performance as measured by PLS Predict was satisfactory; and (f) the model fit was <0.08 in terms of standardized root squared residual (SRMR) values. If all model quality data conformed with the prespecified criteria, we conducted a complete PLS-SEM pathway analysis with 5000 bootstrap samples and calculated the path coefficients (with *p*-values), as well as specific and total indirect (mediated) effects and total effects. The primary statistical analyses were the results of multiple regression analyses, particularly those conducted with PLS-SEM.

## 3. Results

### 3.1. Socio-Demographic Data

We employed a two-step cluster analysis to classify the patients with long COVID into two groups using the peak BT, SpO2, and the NT index (IL-1β + IL-18 + caspase 1+ MPO + AOPP + CRP) with the aim of developing a new biomarker-derived endophenotype class within the long COVID patient group. The diagnosis of long COVID and the peak BT, lowered SpO2, and NT composite score were entered into the cluster analysis as the nominal variable and the continuous variables, respectively. According to the silhouette measure of cohesion and separation of 0.53, the quality of the clusters was adequate. Three clusters were derived; namely, the healthy control sample (n = 39); patients with high peak BT, lowered SpO2, and an increased NT index (dubbed high TO-NT—T for temperature, O for SpO2, and NT for neurotoxicity) (n = 30); and patients (n = 56) with less pronounced changes in these biomarkers (dubbed low TO-NT). Therefore, long COVID patients were classified based on the combination of two acute COVID-19 phase markers and blood biomarkers 3–4 months after clinical recovery.

Socio-demographic and clinical data for the three groups are presented in Table 1. SpO2, peak BT, the TO_2_ index, and the NT index were significantly different between the high and low TO-NT clusters. No significant changes were detected between these groups in terms of age, sex, BMI, marital state, smoking status, residency status, vaccination state, or education.

### 3.2. Differences in Psychiatric Rating Scales between Study Groups

Table 2 shows the measurements of the rating scales; namely, the total and subdomain scores. There were significant differences among the three study groups in total and pure HAMD, HAMA, BDI, and FF and physiosom HAMD and HAMA scores. Table 2 shows that the scores of all scales and subscales, except pure HAMA, increased from controls to the low TO-NT group to the high TO-NT group. The pure HAMA score was not significantly different between the two patient groups but was higher in the latter than in controls.

### 3.3. Differences in Biomarkers between Study Groups

The measurements of the biomarkers in both classes of patients with long COVID versus healthy controls are displayed in Table 3. CRP and AOPP were significantly different between the three study groups. Caspase 1 was significantly higher in the high TO-NT class than in the two other groups. There was a trend towards higher IL-1β levels in the high TO-NT group than in controls and significantly higher IL-10 in both long COVID groups than in controls. Total Ca was lower in patients than in controls.

### 3.4. Prediction of the Physio-Affective Phenome Using TO_2_, NT, and Total Ca

Table 4 shows multiple regression analyses with the physio-affective phenome as the dependent variable and TO_2_, NT, and total Ca as explanatory variables. In regressions #1 and #2, we introduced the first PC extracted from the pure FF, HAMD, HAMA, and BDI and physiosom HAMD and HAMA scores (dubbed the physio-affective phenome PC score, reflecting overall severity) as the dependent variable. We found that 46.0% of the variance in this PC (regression #1) was explained by the total Ca, the NT index, and BMI. Figure 1 and Figure 2 show the partial regression of the physio-affective PC score for the total Ca and the NT index, respectively. After introducing peak BT and SpO2 in regression #2, we found that a large part of the variance (52.4%) in the physio-affective phenome PC was explained by peak BT and NT (both positively associated). Figure 3 shows the partial regression of the phenome score for peak BT. In all regression analyses performed with the pure FF, HAMD, HAMA, and BDI scores and the physiosom HAMD and HAMA scores, the NT index and total Ca were always the most significant predictors.

In order to detect whether a common factor extracted from the NT and TO_2_ indexes and the total Ca was associated with a common factor extracted from the clinical scales, we performed canonical correlation analyses. Table 5 shows the results of this analysis, with the clinical scales as the set of dependent variables and NT, TO_2_, and calcium as the set of explanatory variables. The canonical component extracted from the NT composite, TO_2_, and total Ca strongly correlated with physio-affective symptoms and explained 35.5% of the variance in the latter.

### 3.5. Prediction of the Physio-Somatic and Affective Domains Using Biomarkers

In Table 4, regressions #3 to #6 show the outcomes of regressions with the pure FF, HAMD, HAMA, and BDI scores and the physiosom HAMD and HAMA scores as dependent variables and the separate biomarkers as explanatory variables (without entering the NT index) in order to elucidate which biomarkers were the most predictive. In these regression analyses, we also entered vaccination status as a dummy variable; namely, AstraZeneca (yes = 1, no = 0), Pfizer (yes = 1, no = 0), and Sinopharm (yes = 1, no = 0). We found that 37.1% of the variance in pure FF scores (regression #3) could be explained by total Ca (inversely), CRP, education, AOPP, BMI, and vaccination with AstraZeneca (all positively associated). The results of regression #4 revealed that total Ca (inversely), CRP, education, AOPP, MPO, and vaccination with AstraZeneca (all positively) predicted 41.3% of the variance in pure HAMD scores. We found that, for regression #5, total Ca (inversely associated), CRP, education, AOPP, and IL-1β (all positively associated) explained 37.2% of the variance in the pure BDI score. Regression #6 showed that a significant part of the variance (18.3%) in the pure HAMA score could be predicted by total Ca (inversed associated), CRP, and MPO (both positively associated). The results of regression #7 indicated that, in long COVID patients, 37.4% of the variance in physiosom HAMD scores was explained by total Ca and vaccination with Sinopharm (inversely) and CRP and IL-18 (both positively associated). Regression #8 showed that total Ca (inversely associated), MPO, and BMI (both positively associated) explained 21.7% of the variance in physiosom HAMA scores.

### 3.6. Results of PLS Analysis

Figure 4 depicts the first PLS model, which evaluated whether the effects of SpO2 and peak BT (introduced as a single indicator; namely, the TO_2_ index) on the physio-affective phenome of long COVID (entered as a latent vector taken from the six rating-scale subdomains) were mediated via NT and Ca (IL-10 and TAC were not significant). Since the multiple regression analysis also showed effects of vaccination, we entered vaccination with Sinopharm (yes = 1, no = 0) as an additional explanatory variable. With an SRMR of 0.045, the model quality was satisfactory, and we observed appropriate construct reliability validity values for the physio-affective phenome with AVE = 0.613, rho A = 0.920, composite reliability = 0.904, and Cronbach’s alpha = 0.873. All loadings for the six indicators of the physio-affective phenome were > 0.7. CTA showed that the latter vector was not mis-specified as a reflective model, and blindfolding indicated an acceptable construct cross-validated redundancy of 0.364. PLSPredict showed that the construct indicators had positive Q2 predict values, indicating that the prediction error was lower than the most naive benchmark. Complete PLS path analysis showed that 61.6% of the variance in the physio-affective phenome was explained by the regression for NT, Ca, TO_2_ index, and vaccination and that the TO_2_ index explained 16.2% and 17.1% of the variances in NT and Ca, respectively. SARS-CoV-2 infection explained 47.0% of the variance in the TO_2_ index. While TO_2_ had significant direct effects on the phenome, it also had significant and specific indirect effects mediated via either NT (t = 4.10, *p* < 0.001) or Ca (t = 4.13, *p* < 0.001). Infection resulted in a highly significant total indirect effect on the long COVID phenome (t = 8.92, *p* < 0.001).

In order to examine the effects of SpO2 and peak BT on the separate biomarkers of long COVID and determine which biomarkers are the most important in predicting the phenome, we conducted a second PLS path analysis (see Figure 5). With an SRMR of 0.040, the model quality was adequate, and the construct reliability validity of the latent construct was also adequate (not shown, as it was similar to that explained in Figure 4). We found that 46.8% of the variance was explained by the regression with Ca, CRP, IL-1β, AOPP, MPO, and vaccination. SpO2 had significant effects on AOPP and MPO, whilst peak BT affected MPO, CRP, and Ca.

## 4. Discussion

### 4.1. The Physio-Affective Phenome of Long COVID

The first major outcome of this study was that we were able to extract one replicable latent vector from the physio-somatic and affective rating scale scores. This confirmed the results of another study conducted on an independent sample of Iraqi COVID-19 patients and controls [32]. Moreover, both the latter and the current study found that the physio-affective core of long COVID was strongly predicted by the combined effects of increased peak BT and lowered SpO2 during the acute phase of the disease. As explained previously [31], increased peak BT and lowered SpO2 reflect the severity of the infection–immune–inflammatory core of acute COVID-19. These results indicate that the physio-affective core during acute and long COVID is largely the consequence of infection–immune–inflammatory pathways. The results confirm that physio-somatic symptoms, including chronic fatigue; physio-somatic symptoms, including pain, GIS, malaise, and autonomic symptoms; and affective symptoms share common immune–inflammatory pathways, as reviewed in the introduction.

### 4.2. Increased NT due to NLRP3 Activation Predicts the Physio-Affective Phenome

The second major outcome was that we were able to construct a new endophenotype class based on increased NT during long COVID and lowered SpO2 and increased peak BT during the acute phase of the illness (this cluster was dubbed TO-NT long COVID). The latter cluster of patients was characterized by increased indicants of NLRP3 inflammasome activation, with increased IL-1β and caspase 1, mild inflammation with increased CRP, increased MPO and AOPPs, and lower total Ca levels. It should be underscored that the non-TO-NT cluster of patients also showed increased NT, although significantly less than the TO-NT cluster. Although IL-10, a negative immunoregulatory cytokine, was significantly increased in long COVID, it did not predict the phenome after taking into account the other biomarkers. Most importantly, patients belonging to the TO-NT cluster showed highly significant increases in physio-affective scores, indicating strong associations between biomarkers of acute and long COVID and physio-affective symptoms. Moreover, the NT index and the key components of this index—namely, increased IL-1β, CRP, MPO, and AOPP and lowered Ca—predicted a large part of the variance in the physio-affective phenome. As such, this study has abstracted the physio-affective phenome of long COVID into a more concrete NT-driven concept.

It is interesting to note that NLRP3 genetic variants (namely, NLPR3 rs10157379 T > C and NLPR3 rs10754558 C > G variants) are associated with fatigue, myalgia, hyperalgesia, and malaise in the acute infectious phase [20]. Abnormal NLRP3 activation during acute infection may lead to pathological tissue injury [22] and may underpin the exaggerated immune response, since it contributes to the cytokine storm in acute COVID-19 [51,52]. Several studies have reported that caspase 1, IL-1β, and IL-18 are associated with depression, anxiety, and fatigue, indicating the implication of the NLRP3 inflammasome in the pathophysiology of these diseases [53,54,55,56,57]. NLRP3 can activate the caspase 1 enzyme, which in turn triggers IL-1β and IL-18 pro-inflammatory cytokines to induce pyroptosis (cell death in response to pro-inflammatory signals), and it plays a key role in neuroinflammation [58].

IL-1β is necessary to start and maintain the immune–inflammatory reactions in the central nervous system (CNS) and may impact the integrity of the blood–brain barrier (BBB), leading to leakage of peripheral immune cells into the CNS [59,60]. Moreover, IL-1β mediates microglia and astrocyte activation, resulting in infiltration of T cells into the CNS, thus augmenting the pro-inflammatory state by producing IL-6 and TNF-α along with neurotoxic metabolites, enhancing excitotoxicity and neuronal damage [36,61]. IL-18 is one of the mediators of cell-mediated immunity and triggers T helper (Th) 1 and B cells to generate adhesion molecules, pro-inflammatory cytokines, and chemokines [62,63]. In addition, IL-18 may increase microglial expression of caspase 1 and matrix metalloproteinases and the formation of pro-inflammatory cytokines [64], and it may cause neuronal damage through elevations in Fas ligands in glial cells [57]. High IL-1β and IL-18 levels were detected in patients with CNS infection, brain injuries, Alzheimer’s disease, and multiple sclerosis [65,66,67]. All in all, activation of NLRP3 results in neurotoxic effects through either promoting the synthesis of other detrimental metabolites or damaging neurons directly.

### 4.3. Increased NT due to Increased CRP Predicts the Physio-Affective Core

In accordance with our previous study [32], we found that mild elevations in CRP contributed to the physio-affective phenome of long COVID. Previous studies showed elevated CRP in COVID-19 patients following 2–3 months of full recovery [68,69]. In the liver, CRP production is triggered by IL-6 [70], and elevated CRP has toxic effects on endothelial cells and raises the permeability of the BBB, thus enhancing the development of neurodegenerative and cerebrovascular diseases [71,72,73,74,75]. For example, elevated CRP is associated with impaired functional outcomes and increased mortality due to stroke [76,77,78,79]. Furthermore, studies have shown increased CRP levels related to depression [80,81], suicidal behavior [82], GAD [83], CFS [84], and cognitive impairment [85].

### 4.4. Increased NT due to Oxidative Stress Predicts the Physio-Affective Phenome

The third major finding of the current study was that the physio-affective phenome was associated with increased oxidative stress, as indicated by increased MPO and AOPP; however, in contrast to our prior hypothesis, TAC was not decreased in long COVID. The present findings extend our previous results, revealing high nitro-oxidative stress and lowered antioxidant defenses (lowered zinc and Gpx levels) in long COVID patients [32].

Neutrophils produce MPO enzyme as part of the innate immune response, which may induce the generation of reactive chlorine species (RCS), such as hydrochlorous acid [86], and this can lead to chlorinative stress with the formation of AOPPs [87,88]. In addition to neutrophils, microglia and pyramidal neurons of the hippocampus also express a substantial amount of MPO enzyme, which is associated with disease conditions such as Alzheimer’s disease and multiple sclerosis [89,90]. Increased MPO and AOPP have been reported in relation to depression, anxiety, and cognitive impairment [91,92,93,94].

Moreover, we observed that lowered SpO2 in the acute phase predicted increased AOPP and MPO in long COVID and that increased peak BT in the acute phase predicted increased MPO and CRP and lowered Ca in long COVID. Likewise, our previous results [32] revealed that lowered SpO2 in the acute phase predicted lowered Gpx and increased NO production in long COVID patients and that elevated BT during acute COVID-19 predicted increased CRP and lowered antioxidant defenses, including zinc, in long COVID. This indicates that the effects of the infection–immune–inflammatory core of acute COVID-19 on the physio-affective phenome of long COVID are partly mediated by the cumulative effects of neuro-immune, neuro-oxidative, and neuro-nitrosative pathways.

### 4.5. Lowered Total Ca Levels Predict the Physio-Affective Phenome

The fourth major finding of the current study was that total Ca levels were lower in long COVID patients than in controls and that lowered Ca significantly predicted the severity of the physio-affective phenome. In the acute COVID phase, there is a reduction in Ca that is strongly associated with the physio-affective core [31]. Previous studies have reported that MDD and depressive symptoms are accompanied by significant reductions in ionized and total Ca levels [95,96,97,98,99]. Ca levels are necessary to maintain normal mood and cognition, which occurs through effects on neuronal signaling pathways and protection of neuroplasticity processes [100,101,102]. Moreover, the lowered Ca levels in acute COVID-19 are part of the infection–immune–inflammatory core [31], which also appears to result in lower Ca levels during long COVID. Thus, it is safe to hypothesize that, in patients with long COVID, abnormal Ca, which accompanies the immune–inflammatory response during the acute phase of illness, contributes mechanistically to the physio-affective phenome of both acute and long COVID.

### 4.6. Limitations

The findings of the current study should be interpreted in light of its limitations. First, the results would be more interesting if we had assessed biomarkers of chlorinative stress—namely, chlorotyrosine and dichlorotyrosine—and the activation of the tryptophan catabolite (TRYCAT) pathway, which may increase TRYCATs and lower the availability of tryptophan, the precursor of 5-HT [103,104]. Moreover, our COVID-19 studies were conducted on Iraqi patients and, therefore, require replication in other countries. It would have been more interesting if we had included another study group of patients who were previously infected but showed no symptoms of long COVID in order to compare this cohort with healthy controls. Studies are needed to see whether any of the existing antidepressants or other medications with anti-inflammatory action may be used to treat long COVID. Indeed, antidepressants of different classes suppress the production of interferon γ and increase that of IL-10, thus showing negative immunoregulatory effects [105]. On the other hand, our recent data show that antidepressants, similarly to paroxetine, destroy part of the compensatory immune-regulatory system and, therefore, may worsen the phenome of long COVID (Maes et al., to be submitted).

A more explorative finding is that prior vaccinations with AstraZeneca (viral-vector and genetically modified virus vaccine) or Pfizer (mRNA vaccine) significantly increased the physio-affective phenome scores compared to Sinopharm (inactivated virus vaccine), albeit with a small effect size. As such, these results replicate our previous findings obtained with another Iraqi study sample showing that these vaccinations have a significant effect on the physio-somatic symptoms of the HAMA and HAMD [32]. Moreover, these vaccinations are known to induce long COVID-like symptoms, including depression, fatigue, and anxiety, in association with increased spike protein synthesis, type 1 interferon signaling, T cell activation, and autoimmune responses [106,107]. The statistical association between these vaccination types and long COVID physio-somatic symptoms was not a primary outcome result but, in fact, an explorative finding; therefore, these results should be re-examined in prospective cohort studies.

It is also possible that different variants are associated with different long COVID symptoms, with the Alpha variant possibly exhibiting more mental health and cognitive symptoms than the original variant [108]. A few months (3–4) prior to the September–December 2021 recruitment of our long COVID patients, the Alpha variant (B.1.1.7) of SARS-CoV-2 was predominant in Iraq (80.2%), while only one B.1.351 strain was detected [109]. Around 29 September 2021, the first Omicron variants appeared [110]. Therefore, in the current study, we mainly evaluated the effects of acute infection with the B.1.1.7 Alpha variant and, to a lesser extent, the B.1.351 variant on long COVID 3–4 months after the acute infection. Examining the effects of other variants on the immune–inflammatory processes during acute infection and the neurotoxicity and decreased total Ca during long COVID would be extremely interesting.

## 5. Conclusions

High peak BT and lowered SpO2 during the acute phase of COVID-19 are associated with the development of the physio-affective phenome of long COVID disease, and these effects are partially explained by increased NT through activation of the NLRP3 inflammasome, a mild inflammatory response, increased chlorinative stress, and lowered total Ca levels. The infection–immune–inflammatory core of acute COVID-19, the development of NLRP3, a mild chronic inflammatory response, increased nitro-oxidative stress, lowered antioxidant defenses, and total Ca levels can be considered new drug targets to treat the physio-affective symptoms of long COVID.

## Figures and Tables

**Figure 1 jcm-12-00511-f001:**
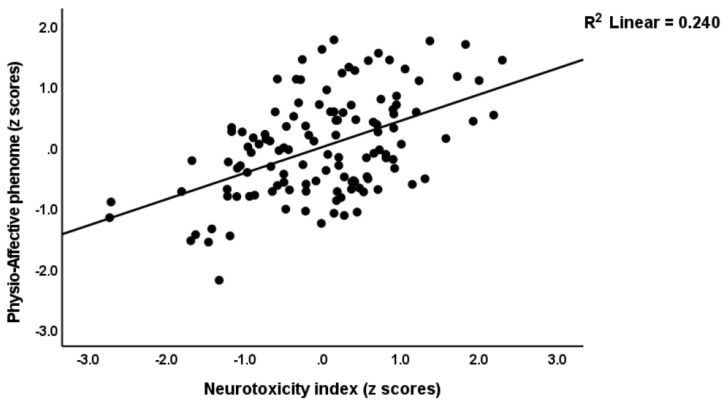
Partial regression for the physio-affective phenome score in long COVID patients and healthy controls against the neurotoxicity index (IL-1β + IL-18 + caspase 1 + MPO + AOPP + CRP).

**Figure 2 jcm-12-00511-f002:**
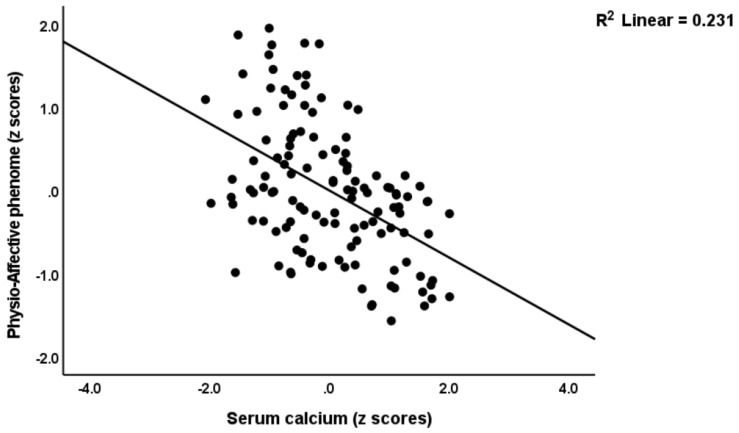
Partial regression for the physio-affective phenome score in long COVID patients and healthy controls against serum total calcium.

**Figure 3 jcm-12-00511-f003:**
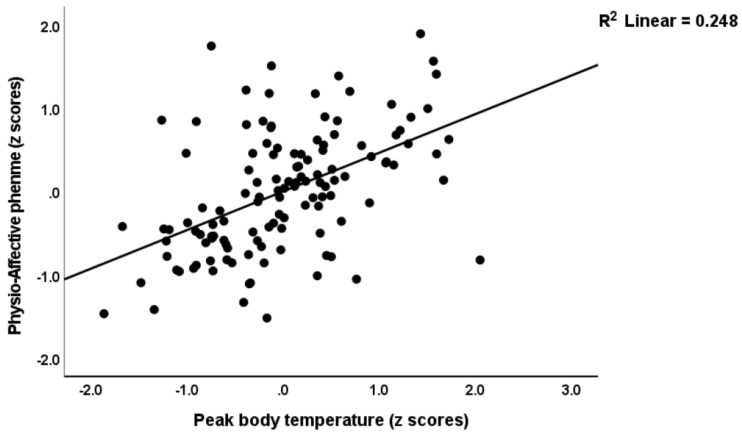
Partial regression for the physio-affective phenome score in long COVID patients and healthy controls on peak body temperature.

**Figure 4 jcm-12-00511-f004:**
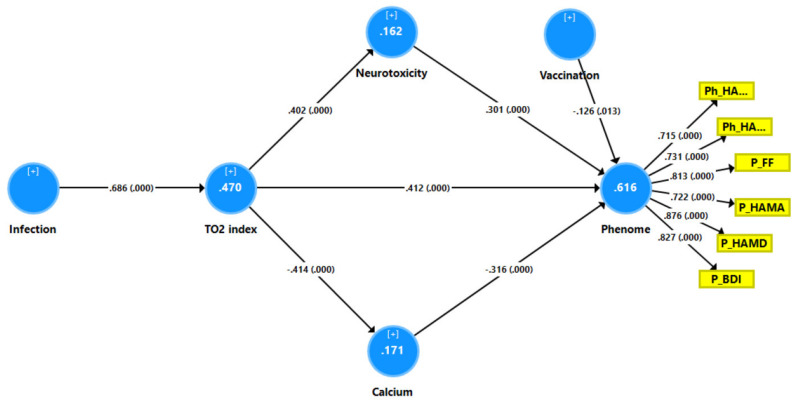
Results of partial least squares (PLS)–SEM analysis with the physio-affective phenome score as the output variable. The latter was entered as a latent vector (blue circle) extracted from six symptom domains (yellow shapes). The phenome latent vector was predicted by calcium, neurotoxicity index, the TO_2_ index, and vaccination (AstraZeneca and Pfizer were coded as 0, Sinopharm as 1). SARS-CoV-2 infection was the primary input variable. As such, neurotoxicity and calcium partially mediated the effects of TO_2_ and SARS-CoV-2 infection on the physio-affective phenome. Neurotoxicity: z unit-based composite score computed based on six neurotoxic immune and oxidative products. TO_2_: z unit-based composite score computed as z BT—z SpO2. Vaccination: indicates that AstraZeneca and Pfizer vaccinations were positively associated with the phenome. PhHAMA: physio-somatic symptoms of the Hamilton Anxiety Rating Scale; PhHAMD: physio-somatic symptoms of the Hamilton Depression Rating Scale; PuFF: pure fatigue and physio-somatic symptoms of the Fibro Fatigue scale; PuHAMA: pure anxiety symptoms of the Hamilton Anxiety Rating Scale; PuHAMD: pure depression symptoms of the Hamilton Depression Rating Scale; PuBDI: pure depression scores from the Beck Depression Inventory. Vaccination: Sinopharm was entered as a dummy variable (yes = 1, no = 0).

**Figure 5 jcm-12-00511-f005:**
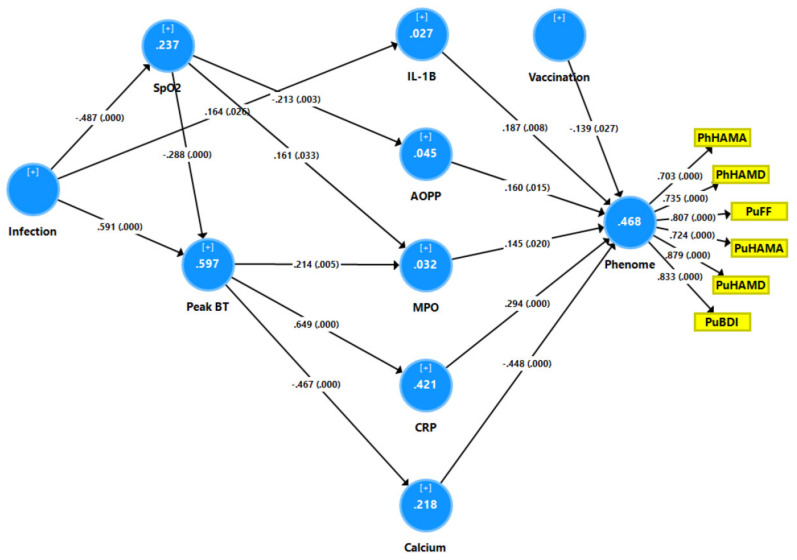
Results of partial least squares (PLS)–SEM analysis with the physio-affective phenome score as the output variable. The latter was entered as a latent vector (blue circle) extracted from six symptom domains (yellow shapes). The phenome latent vector was predicted by calcium, all neurotoxicity biomarkers (namely, interleukin (IL) 1β, C-reactive protein (CRP), myeloperoxidase (MPO), and advanced oxidation protein products), peak body temperature (BT), oxygen saturation (SpO2), and vaccination (AstraZeneca and Pfizer were coded as 0, Sinopharm as 1). SARS-CoV-2 infection was the primary input variable. As such, the neurotoxicity biomarkers and calcium mediated the effects of BT, SpO2, and infection on the phenome. PhHAMA: physio-somatic symptoms of the Hamilton Anxiety Rating Scale; PhHAMD: physio-somatic symptoms of the Hamilton Depression Rating Scale; PuFF: pure fatigue and physio-somatic symptoms of the Fibro Fatigue scale; PuHAMA: pure anxiety symptoms of the Hamilton Anxiety Rating Scale; PuHAMD: pure depression symptoms of the Hamilton Depression Rating Scale; PuBDI: pure depression scores from the Beck Depression Inventory. Vaccination: Sinopharm was entered as a dummy variable (yes = 1, no = 0).

**Table 1 jcm-12-00511-t001:** Socio-demographic data, body temperature (BT), and oxygen saturation (SpO2) in healthy controls (HC) and long COVID patients classified according to peak body temperature (BT), oxygen saturation (SpO2), and neurotoxicity (NT) index as long COVID with high (high TO-NT) versus low (low TO-NT) levels of these biomarkers.

Variables	HC (n = 39) ^A^	Low TO-NT (n = 56) ^B^	High TO-NT (n = 30) ^C^	F/X^2^	df	*p*
Age (years)	28.3 (7.6)	27.6 (5.4)	29.8 (7.3)	1.15	2/122	0.320
Sex (M/F)	42/15	40/16	22/8	1.42	2	0.512
Marital state (Ma/S)	22/17	31/25	18/12	0.18	2	0.946
Smoking (Y/N)	13/26	16/40	11/19	0.64	2	0.732
Residency (U/R)	31/8	46/10	25/5	0.19	2	0.914
Vaccination (A/Pf/S)	9/21/9	14/31/11	6/17/7	0.42	4	0.979
BMI (Kg/m^2^)	25.60 (3.97)	25.71 (3.74)	26.96 (5.75)	0.99	2/122	0.372
Education (years)	15.0 (1.3)	15.7 (1.8)	15.5 (1.7)	2.73	2/122	0.069
Peak BT (℃)	36.86 (0.25) ^B,C^	38.20 (0.64) ^A,C^	39.31 (0.83) ^A,B^	138.38	2/122	<0.001
Lowest SpO2 (%)	95.08 (1.52) ^B,C^	92.41 (2.74) ^A,C^	88.27 (4.62) ^A,B^	42.81	2/122	<0.001
TO_2_ index (z scores)	−1.01 (0.240) ^B,C^	0.054 (0.52) ^A,C^	1.21 (0.86) ^A,B^	131.43	2/122	<0.001
Composite NT (z score)	−0.595 (0.85) ^B,C^	−0.143 (0.80) ^A,C^	1.04 (0.66) ^A,B^	37.91	2/122	<0.001

Results are shown as means (SD). F: results of analysis of variance; X^2^: analysis of contingency tables, df: degrees of freedom. ^A,B,C^: Results of pairwise comparisons among group means. TO-NT: (T) temperature, (O) oxygen saturation, (NT) neurotoxic index. M: male, F: female, Ma: married, S: single, U: urban, R: rural, BMI: body mass index, A: AstraZeneca, Pf: Pfizer, S: Sinopharm, Y: yes, N: no. TO_2_ index computed as z BT—z SpO2; neurotoxicity (NT) computed as a z unit-based composite score.

**Table 2 jcm-12-00511-t002:** Clinical rating scale scores in healthy controls (HC) and long COVID patients classified according to peak body temperature (BT), oxygen saturation (SpO2), and neurotoxicity (NT) index as long COVID with high versus low levels of these biomarkers (dubbed TO-NT).

Variables	HC (n = 39) ^A^	Low TO-NT (n = 56) ^B^	High TO-NT (n = 30) ^C^	F/X^2^ df = 2/119	*p*
Total HAMD	5.5 (0.67) ^B,C^	15.7 (0.6) ^A,C^	19.1 (0.8) ^A,B^	105.21	<0.001
Total BDI	8.6 (1.0) ^B,C^	23.5 (0.9) ^A,C^	27.0 (1.2) ^A,B^	85.35	<0.001
Total HAMA	7.8 (1.1) ^B,C^	15.4 (0.9) ^A,C^	19.5 (1.3) ^A,B^	26.23	<0.001
Total FF	10.9 (1.8) ^B,C^	24.7 (1.5) ^A,C^	35.4 (2.0) ^A,B^	43.05	<0.001
Pure FF (z score)	−0.877 (0.122) ^B,C^	0.146 (0.101) ^A,C^	0.867 (0.138) ^A,B^	46.06	<0.001
Pure HAMD (z score)	−0.964 (0.118) ^B,C^	0.296 (0.098) ^A,C^	0.702 (0.133) ^A,B^	51.04	<0.001
Pure BDI (z score)	−1.030 (0.113) ^B,C^	0.348 (0.094) ^A,C^	0.700 (0.128) ^A,B^	62.79	<0.001
Pure HAMA (z score)	−0.606 (0.139) ^B,C^	0.190 (0.115) ^A^	0.432 (0.157) ^A^	14.37	<0.001
Physiosom HAMA (z score)	−0.495 (0.145) ^B,C^	0.029 (0.120) ^A,C^	0.588 (0.164) ^A,B^	12.21	<0.001
Physiosom HAMD (z score)	−1.031 (0.118) ^B,C^	0.347 (0.098) ^A,C^	0.702 (0.133) ^A,B^	57.93	<0.001

All results of univariate GLM analysis; data are expressed as estimated marginal mean (SE) values obtained by GLM analysis after covarying for age, sex, education, and smoking. ^A,B,C^: Results of pairwise comparisons among group means. FF: Fibro Fatigue scale, HAMA: Hamilton Anxiety Rating Scale, HAMD: Hamilton Depression Rating Scale, BDI: Beck Depression Inventory II, Physiosom: physio-somatic.

**Table 3 jcm-12-00511-t003:** Biomarkers in healthy controls (HC) and long COVID patients classified according to peak body temperature (BT), oxygen saturation (SpO2), and neurotoxicity (NT) index as long COVID with high (high TO-NT) versus low (low TO-NT) levels of these biomarkers.

Biomarkers	HC (n = 39) ^A^	Low TO-NT (n = 56) ^B^	High TO-NT (n = 30) ^C^	F/X^2^ df = 2/117	*p*
Caspase 1 (pg/mL)	73.90 (3.27) ^C^	71.83(2.75) ^C^	85.57(3.75) ^A,B^	4.54	0.013
IL-1β (pg/mL)	4.58 (0.33) ^C^	5.21(0.282)	5.81(0.385) ^A^	2.95	0.056
IL-18 (pg/mL)	233.9 (11.91)	231.62(10.02)	261.32(13.67)	1.67	0.192
IL-10 (pg/mL)	9.09 (1.08) ^B,C^	14.10(0.911) ^A^	13.06(1.24) ^A^	6.50	0.002
CRP (mg/L)	5.02 (0.53) ^B,C^	6.32 (0.443) ^A,C^	10.11 (0.604) ^A,B^	27.08	<0.001
MPO (ng/mL)	43.1 (3.3)	49.9 (2.8)	51.3 (3.8)	1.72	0.184
TAC (U/mL)	6.74 (0.51)	6.78 (0.43)	7.02 (0.58)	0.08	0.925
AOPP (µmol/g)	0.92 (0.14) ^B,C^	1.29 (0.12) ^A,C^	1.76 (0.16) ^B,C^	7.80	0.001
Total calcium (mM)	2.56 (0.03) ^B,C^	2.26 (0.02) ^A^	2.33 (0.03) ^A^	40.45	<0.001

All results of univariate GLM analysis; data are expressed as estimated marginal mean (SE) values obtained by GLM analysis after covarying for age, sex, education, and smoking. ^A,B,C^: Results of pairwise comparisons among group means. IL: interleukin, CRP: C-reactive protein, MPO: myeloperoxidase, TAC: total antioxidant capacity, AOPP: advanced oxidation protein product.

**Table 4 jcm-12-00511-t004:** Results of multiple regression analyses with different physio-somatic and affective rating scale scores as dependent variables and peak body temperature (BT), oxygen saturation (SpO2), neurotoxicity (NT), and biomarkers as explanatory variables.

Dependent Variables	Explanatory Variables	B	t	*p*	Model R^2^	F	df	*p*
#1 Physio-somatic phenome	Model Total calcium NT BMI	−0.414 0.420 0.164	−6.00 6.16 2.42	<0.001 <0.001 0.017	0.460	34.06	3/120	<0.001
#2 Physio-somatic phenome	Model Peak BT Calcium NT	0.472 −0.253 0.229	6.29 −3.76 3.38	<0.001 <0.001 <0.001	0.574	53.88	3/120	<0.001
#3 Pure FF	Model Total calcium CRP Education AOPP AstraZeneca vaccination BMI	−0.346 0.242 0.192 0.214 0.166 0.149	−4.56 3.09 2.59 2.74 2.23 1.99	<0.001 0.002 0.011 0.007 0.027 0.048	0.371	11.61	6/118	<0.001
#4 Pure HAMD	Model Total calcium CRP Education AstraZeneca vaccination AOPP MPO	−0.381 0.224 0.244 0.186 0.201 0.155	−5.25 2.97 3.37 2.58 2.67 2.11	<0.001 0.004 0.001 0.011 0.009 0.037	0.413	13.86	6/118	<0.001
#5 Pure BDI	Model Total calcium AOPP Education CRP Interleukin-1β	−0.368 0.247 0.197 0.218 0.174	−4.87 3.17 2.61 2.80 2.22	<0.001 0.002 0.010 0.006 0.028	0.372	14.12	5/119	<0.001
#6 Pure HAMA	Model Total calcium CRP MPO	−0.296 0.204 0.170	−3.52 2.42 2.06	0.001 0.017 0.041	0.183	9.06	3/121	<0.001
#7 Physiosom HAMD	Model Total calcium CRP Interleukin 18 Vaccination Sinopharm	−0.432 0.301 0.181 −0.159	−5.84 4.07 2.49 −2.19	<0.001 <0.001 0.014 0.030	0.374	17.88	4/120	<0.001
#8 Physiosom HAMA	Model Total calcium MPO BMI	−0.301 0.231 0.207	−3.70 2.85 2.53	<0.001 0.005 0.013	0.217	11.17	3/121	<0.001

FF: Fibro Fatigue scale, HAMA: Hamilton Anxiety Rating Scale, HAMD: Hamilton Depression Rating Scale, BDI: Beck Depression Inventory, Physiosom: physio-somatic, BMI: body mass index, CRP: C-reactive protein, AOPP: advanced oxidation protein product, MPO: myeloperoxidase. Vaccinations: entered as dummy variables (yes = 1, no = 0).

**Table 5 jcm-12-00511-t005:** Results of canonical correlation analyses examining the associations between two sets of variables; namely, set 1: the physio-somatic and affective rating scale scores as dependent variables, and set 2: neurotoxicity (NT), a composite based on peak body temperature and SpO2 (TO_2_ index), and serum calcium as explanatory variables.

Sets	Variables	Canonical Loadings
Set 1 Dependent	Pure FF	0.845
	Pure HAMD	0.815
	Physiosom HAMD	0.784
	Pure HAMA	0.585
	Physiosom HAMA	0.629
	Pure BDI	0.825
Set 2 Explanatory	NT composite	0.670
	TO_2_ index	0.875
	Total calcium	0.674
Statistics	F (df)	8.675 (18/325)
	P	<0.001
	Correlation	0.799
	Set 1/set 2	0.355
	Set 2 by itself	0.556
	Set 1 by itself	0.569

FF: Fibro Fatigue scale, HAMA: Hamilton Anxiety Rating Scale, HAMD: Hamilton Depression Rating Scale, BDI: Beck Depression Inventory, Physiosom: physio-somatic, NT: a z unit-based composite score reflecting neurotoxicity; TO_2_: computed as z peak BT—SpO2.

## Data Availability

The dataset generated and/or analyzed during the current study is available from the corresponding author (M.M.) upon reasonable request and once the dataset has been fully exploited by the authors.

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
