# Peer review of "Chronic Fatigue, Depression and Anxiety Symptoms in Long COVID Are Strongly Predicted by Neuroimmune and Neuro-Oxidative Pathways Which Are Caused by the Inflammation during Acute Infection"

_jcm, 2023, doi:10.3390/jcm12020511_

Round 1

Reviewer 1 Report

The aims of this study are to demonstrate that body temperature and SpO2 during acute COVID-19 predict the development of physio-affective symptoms in the following months, and that this association is likely due to a persistent inflammatory reaction, whose features are an increase in blood concentration of IL-1β, IL-18, IL-10, caspase-1, MPO, TAC, AOPP, and a reduction of serum Ca.

I suggest those minimal corrections:

-at line 118 “patients should have had a confirmed infection with COVID-19”

-at line 120 “and be present 3-4 months after recovery”

I agree with the author that the lack of a study group of patients who were previously infected but did not show any symptoms of Long COVID is a significant limitation. Anyway, with this explicit limit, I do not have found other critical elements.

Author Response

COVID-19 predict the development of physio-affective symptoms in the following months, and that this association is likely due to a persistent inflammatory reaction, whose features are an increase in blood concentration of IL-1β, IL-18, IL-10, caspase-1, MPO, TAC, AOPP, and a reduction of serum Ca.

I suggest those minimal corrections:

-at line 118 “patients should have had a confirmed infection with COVID-19”

@ANSWER: Now changed as the reviewer suggested

-at line 120 “and be present 3-4 months after recovery”

@ANSWER: Now corrected

I agree with the author that the lack of a study group of patients who were previously infected but did not show any symptoms of Long COVID is a significant limitation. Anyway, with this explicit limit, I do not have found other critical elements.

Reviewer 2 Report

Dear Authors,

I read with interest your work entitled ” Chronic fatigue, depression and anxiety symptoms in Long COVID are strongly predicted by neuroimmune and neuro-oxidative pathways which are caused by the inflammation during acute infection”. It’s a very ambitious project and fascinating, but I have some suggestions. First, I appreciate the considerable work to realize all the statistical analysis you have done, but I think It’s very hard for a reader to understand and complete the lecture of the article. This aspect influence the complete comprehension of the results of your study. The excessive data manipulation confuses the readers, losing the aim of the study. On the other hand, the discussion and the conclusion are really weak compared to the statical analysis.

Moreover, self-citation should be reduced; It’s not ethically correct. So I invite you to write the article again, trying to simplify the “ materials and methods” and “results” sections to make the work more comprehensive. In addition, I suggest submitting as supplementary files the details of your statistical elaboration not necessary to describe the principal aim of the study. Consequently,  It’s helpful to rewrite the “Discussion” and the ” Conclusion” sections.

Furthermore, the following aspects need Minor revision:

·        Line 133: Did you perform immunoglobulin M (IgM) against SARS-COV-2 for all the patients admitted to the hospital? Is it in your guidelines a criterion for defining a positive COVID-19 infection case?

·        Lines 135-137: please write in which nation has located the hospitals

·        Line 187 and 509: It could be better to replace “in our previous studies” by citing the author and the study (es: In the study conducted by Al-Hadrawi et al.…..)

Author Response

Dear Reviewer:

We think our data is helpful to the overall conclusion of the article, so we have made reservations.

However, there are 5/108 to Al-Hakaim, and 13/108 to Maes et al. (i.e. different groups all over the world). We think citations should be used based on priority: one should always cite those authors who were the first to report the findings. Otherwise, it is citation amnesia, widely used by American groups. There is an example at the below link:
https://pubmed.ncbi.nlm.nih.gov/25789583/
In this case, Maes and his groups were always the primary sources. 

The first part of this study was performed on another study sample: 
https://pubmed.ncbi.nlm.nih.gov/36280755/

Other changes can be seen in the revised version. 

Reviewer 3 Report

Long COVID is a very important topic.

page 2, line 58, please add: Available treatments are limited as there is no clear understanding of the pathophysiological mechanism. 

page 16, line 539, please add: Studies are needed to see if any of the existing anti-depressants like Fluvoxamine and other medications have anti-inflammatory action and treat long COVID. 

Author Response

@REVIEWER #3

Long COVID is a very important topic.

page 2, line 58, please add: Available treatments are limited as there is no clear understanding of the pathophysiological mechanism. 

@ANSWER: In the revision we added (line 104):

Available treatments are limited as there is no clear understanding of the pathophysiological mechanism.

page 16, line 539, please add: Studies are needed to see if any of the existing anti-depressants like Fluvoxamine and other medications have anti-inflammatory action and treat long COVID. 

@ANSWER: addressed in the text as:

Studies are needed to see if any of the existing antidepressants and other medications with anti-inflammatory action may be used to treat Long COVID. Indeed antidepressants of different classes suppress the production of interferon-γ and increase that of IL-10, thus, showing negative immunoregulatory effects [104]. On the other hand, our recent data show that antidepressants like paroxetine destroy part of the compensatory immune-regulatory system and therefore may worsen the phenome of Long COVID (to be submitted).